# Mechanisms Underlying the Suppression of IL-1β Expression by Magnesium Hydroxide Nanoparticles

**DOI:** 10.3390/biomedicines11051291

**Published:** 2023-04-27

**Authors:** Ayaka Koga, Chuencheewit Thongsiri, Daisuke Kudo, Dao Nguyen Duy Phuong, Yoshihito Iwamoto, Wataru Fujii, Yoshie Nagai-Yoshioka, Ryota Yamasaki, Wataru Ariyoshi

**Affiliations:** 1Department of Health Sciences, Kyushu Dental University, Kitakyushu 803-8580, Fukuoka, Japan; r20koga@fa.kyu-dent.ac.jp; 2Division of Infections and Molecular Biology, Department of Health Promotion, Kyushu Dental University, Kitakyushu 803-8580, Fukuoka, Japan; r16yoshioka@fa.kyu-dent.ac.jp (Y.N.-Y.); r18yamasaki@fa.kyu-dent.ac.jp (R.Y.); 3Department of Conservative Dentistry and Prosthodontics, Srinakharinwirot University, Bangkok 10110, Thailand; chuencheewit@g.swu.ac.th; 4SETOLAS Holdings Inc., Sakaide 762-0012, Kagawa, Japan; kudo.daisuke@setolas.co.jp (D.K.); dao.nguyen.duy.phuong@setolas.co.jp (D.N.D.P.); iwamoto.yoshihito@setolas.co.jp (Y.I.); 5Unit of Interdisciplinary Promotion, School of Oral Health Sciences, Faculty of Dentistry, Kyushu Dental University, Kitakyushu 803-8580, Fukuoka, Japan; r15fujii@fa.kyu-dent.ac.jp

**Keywords:** magnesium hydroxide nanoparticles, macrophages, MAPK pathways, NF-κB pathways, innate immunity, dentistry, periodontology

## Abstract

In recent years, magnesium hydroxide has been widely studied due to its bioactivity and biocompatibility. The bactericidal effects of magnesium hydroxide nanoparticles on oral bacteria have also been reported. Therefore, in this study, we investigated the biological effects of magnesium hydroxide nanoparticles on inflammatory responses induced by periodontopathic bacteria. Macrophage-like cells, namely J774.1 cells, were treated with LPS derived from *Aggregatibacter actinomycetemcomitans* and two different sizes of magnesium hydroxide nanoparticles (NM80/NM300) to evaluate their effects on the inflammatory response. Statistical analysis was performed using an unresponsive Student’s t-test or one-way ANOVA followed by Tukey’s post hoc test. NM80 and NM300 inhibited the expression and secretion of IL-1β induced by LPS. Furthermore, IL-1β inhibition by NM80 was dependent on the downregulation of PI3K/Akt-mediated NF-κB activation and the phosphorylation of MAPK molecules such as JNK, ERK1/2, and p38 MAPK. By contrast, only the deactivation of the ERK1/2-mediated signaling cascade is involved in IL-1β suppression by NM300. Although the molecular mechanism involved varied with size, these results suggest that magnesium hydroxide nanoparticles have an anti-inflammatory effect against the etiologic factors of periodontopathic bacteria. These properties of magnesium hydroxide nanoparticles can be applied to dental materials.

## 1. Introduction

Periodontitis is one of the most frequent chronic inflammatory diseases caused by bacterial infections, and it is becoming increasingly prevalent in today’s aging society. Progressive periodontitis results in the destruction of periodontal tissue and alveolar bone resorption, eventually resulting in tooth loss. In addition, periodontitis is associated with systemic diseases, such as diabetes [1], myocardial infarction [2], and aspiration pneumonia [3,4]. In an aging society, lowering periodontitis prevalence is critical for maintaining oral and general health.

Periodontopathic bacteria, such as *Porphyromonas gingivalis*, *Tannerella forsythia*, *Treponema denticola*, and *Aggregatibacter actinomycetemcomitans*, were reported to initiate and progress periodontitis [5,6,7]. Most periodontopathic bacteria are anaerobic Gram-negative bacteria that can be identified locally in periodontal tissues such as periodontal pockets. Periodontopathic bacteria possess several virulent factors that are implicated in various pathologies of periodontitis. Lipopolysaccharide (LPS) is a component of the outer membrane of Gram-negative bacteria and possesses endotoxic activity that induces an inflammatory response by binding to its receptors such as Toll-like receptor 4 (TLR4) [8].

Macrophages are immune cells that are responsible for a rapid response to bacterial infection. Macrophages vary from bone marrow-derived monocytes in that they have several bioactivities that allow them to recognize, internalize, and kill microbial pathogens. The recognition of microbial pathogens, including LPS, by macrophages, activates inflammatory signaling pathways and induces the production of various pro-inflammatory cytokines, such as IL-1β [9,10].

Interleukin-1β (IL-1β) is a proinflammatory cytokine that is required for host defense against infection and injury [11]. Although IL-1β is produced and secreted by a variety of cell types, most studies have focused on its synthesis in innate immune system cells, including monocytes and macrophages [12]. Although the excessive release of IL-1β induces acute inflammation, prolonged local IL-1β production induces the destruction of bones, joints, and blood vessels in chronic inflammatory diseases, such as rheumatoid arthritis [13]. In addition, periodontopathic bacteria also induce IL-1β production in innate immune cells [14,15]. As the expression level of IL-1β in periodontal tissue and gingival cervical fluid is directly associated with the severity of periodontitis [16,17], regulating IL-1β expression is critical in developing a strategy for the prevention and treatment of periodontitis.

The production of pro-inflammatory cytokines in the periodontal tissue causes gingival swelling and edema, gingival attachment loss, and periodontal pocket formation. Pro-inflammatory cytokines also stimulate the expression of the receptor activator of nuclear factor kappa-Β ligand (RANKL) on the surface of osteoblasts [18] and induce osteoclastogenesis, resulting in alveolar bone destruction. Moreover, periodontopathic bacteria have various invasive host defense mechanisms. Capsular periodontopathic bacteria inhibit phagocytosis and produce enzymes that degrade immunoglobulins. These immune evasive systems of periodontopathic bacteria generate hyperinflammatory responses in periodontal pockets. As a result, controlling macrophage-mediated inflammatory reactions can be critical for developing novel therapeutic strategies for inflammatory diseases, such as periodontitis.

Recently, magnesium hydroxide (Mg(OH)_2_) has attracted attention because of its low-cost biocompatibility, bactericidal properties, and clinical application as a laxative and anti-glaucoma agent [19]. Previous studies revealed the antimicrobial activity of Mg(OH)_2_ against *Chlamydomonas reinhardtii*, *Saccharomyces cerevisiae*, and *Escherichia coli* (*E. coli*) [20,21]. Furthermore, we demonstrated that Mg(OH)_2_ nanoparticles (D50 = 75.2 nm) exhibit a physical killing activity against exponential and persistent *E. coli* [22]. In addition, Mg(OH)_2_ nanoparticles suppress inflammatory responses in the porcine coronary artery [23] and mouse renal glomeruli [24]. In this study, we investigated the biological effects of Mg(OH)_2_ on inflammatory responses induced by periodontopathic bacteria in macrophages.

## 2. Materials and Methods

### 2.1. Reagents

Nigericin and anti-β-actin (mouse) monoclonal antibodies were purchased from Sigma-Aldrich (St. Louis, MO, USA). LPS was prepared by *A. actinomycetemcomitans* as described previously [25,26]. LY294002 was purchased from MedChem Express (Monmouth Junction, NJ, USA). The JNK Inhibitor II and PD98059 were purchased from Merck Millipore (Billerica, MA, USA). SB239063 was purchased from Calbiochem (San Diego, CA, USA). Mg(OH)_2_ nanoparticles were prepared in the same way as previously described [20]. Anti-IL-1β (rabbit) monoclonal, anti-IκBα (rabbit) polyclonal, anti-NF-κB p65 (rabbit) monoclonal, anhosphorpho-p38 MAPK (rabbit) monoclonal, anti-p38 MAPK (rabbit) polyclonal, anti-phospho-ERK1/2 (rabbit) monoclonal, anti-ERK1/2 (rabbit) monoclonal, anti-phospho-JNK (rabbit) polyclonal, anti-JNK (rabbit) polyclonal, anti-phospho-Akt (rabbit) monoclonal, and anti-Akt (rabbit) polyclonal antibodies were purchased from Cell Signaling Technology Inc. (Beverly, MA, USA). An anti-β-actin (mouse) monoclonal antibody was purchased from Sigma-Aldrich (St. Louis, MO, USA). The anti-lamin B1 (rabbit) polyclonal antibody was purchased from Proteintech Biotechnologies (Rosement, IL, USA).

### 2.2. Cell Culture

A mouse macrophage cell line, namely J774.1 cells (RCB0434), was purchased from RIKEN CELL BANK (Ibaraki, Japan) and cultured in RPMI 1640 (Wako Pure Chemicals, Osaka, Japan) containing 10% heat-inactivated fetal bovine serum (FBS) (Sigma-Aldrich), 100 units/mL penicillin, and 100 μg/mL streptomycin (Wako Pure Chemicals) [26]. Cells were cultured in an incubator at 37 °C in a 5% CO_2_ atmosphere.

### 2.3. WST-8 Analysis

J774.1 cells (5.0 × 10^4^ cells/well) were cultured in 96-well microplates in 100 μL of 10% FBS in RPMI 1640 containing Mg(OH)_2_ nanoparticles (0, 50, 100, 250, and 500 µg/mL). The cells were cultured at 37 °C in a 5% CO_2_ incubator for 48 h. Cell numbers were determined using a cell counting kit-8 (Dojindo Molecular Technologies, Inc., Rockville, MD, USA). The cells were incubated with CCK-8 (10 μL/well) for 4 h. The absorbance was read at 450 nm using a spectrophotometer (Multiskan FC; Thermo Fisher Scientific, Rockford, IL, USA) [26].

### 2.4. Real-Time RT-qPCR

J774.1 cells (1.0 × 10^6^ cells/well) were seeded in 6-well plates with 10% FBS in RPMI 1640 and stimulated with LPS (2 ng/mL) in the presence or absence of NM80/NM300 for 2 h. In some experiments, the cells were pretreated with inhibitors for 1 h prior to stimulation with LPS. The total RNA was isolated from cultured cells according to the protocol of the Cica Geneus RNA Prep Kit (Kanto Chemical Co., Inc., Tokyo, Japan). The concentration of extracted RNA was measured with a NanoDrop™ 2000/2000c spectrophotometer (Thermo Fisher Scientific), and reverse transcription reactions were performed on a Thermal Gene Atlas (Astec, Fukuoka, Japan) using a ReverTra Ace DPCR RT Master Mix kit (Toyobo Co, Osaka, Japan). PCR products were detected using a qPCR Brilliant III SYBR Master Mix with ROX (Agilent Technologies, Inc., Santa Clara, CA, USA) and an AriaMx Real-Time PCR system (version 1.6; Agilent Technologies). Relative mRNA expression levels were calculated using the 2ΔCt method with GAPDH as the housekeeping gene. The primers used were GAPDH, 5′-GACGGCCGCATCTTCTTGA-3′ (forward) and 5′-CACACACCGACCTTCACCATTTT-3′ (reverse), Il-1β, 5′-AAGGGCTGCTTCCAAACCTTTGAC-3′ (forward), and 5′-ATTGCTTGGGATCCACACTCTCCAACCTTTGAC-3′ (reverse) [26].

### 2.5. Western Blotting

J774.1 cells (1.0 × 10^6^ cells/well) were seeded in 6-well plates with 10% FBS in RPMI 1640 and stimulated with LPS (2 ng/mL) in the presence or absence of NM80/NM300 for 30 min. In some experiments, cells were pretreated with inhibitors for 1 h prior to their stimulation with LPS. Whole-cell lysates were prepared using a Cell Lysis Buffer (Cell Signaling Technology Inc.) supplemented with a protease inhibitor (Thermo Fisher Scientific). In some experiments, nuclear and cytoplasmic fractions were extracted using NE-PER (Thermo Fisher Scientific) according to the manufacturer’s protocol. Total protein concentrations were determined using a DC protein assay kit (Bio-Rad, Hercules, CA, USA). Equal amounts of protein were separated by sodium dodecyl sulfate-polyacrylamide gel electrophoresis (SDS-PAGE) and transferred to polyvinylidene fluoride membranes (Pall Life Sciences, Port Washington, DC, USA). Non-specific binding sites on the membrane were blocked using Blocking One (Nacalai Tesque, Kyoto, Japan) for 30 min at room temperature. Incubation with diluted primary antibodies was performed at 4 °C overnight, followed by incubation with horseradish peroxidase-conjugated secondary antibodies (Amersham™ GE Healthcare) for 1 h at room temperature. Rabbit IgG HRP-linked whole Ab (donkey) and Mouse IgG HRP-linked whole Ab (sheep) was used as secondary antibodies. For signal detection, chemiluminescence was generated using an ECL reagent (Amersham™ GE Healthcare) and detected with a digital system using GelDoc™ XR Plus (version 2.0.1; Bio-Rad) [26].

### 2.6. ELISA

J774.1 cells (1.0 × 10^6^ cells/well) were seeded in 6-well plates, cultured in RPMI 1640, supplemented with 10% FBS, and incubated with LPS (2 ng/mL) in the presence or absence of NM80 or NM300 (50 μg/mL) for 5 h, followed by nigericin (15 μM) stimulation for 1 h. The conditioned medium was collected, and the IL-1β protein concentration was measured by ELISA according to the manufacturer’s instructions. The mouse IL-1β ELISA kit was purchased from R&D Systems (Minneapolis, MN, USA). The secreted protein was measured by spectrophotometry at 450 nm [26].

### 2.7. Statistical Analysis

Data were analyzed using Excel (version 15.0; Microsoft, Redmond, WA, USA) and SPSS software (version 15.0; IBM, Armonk, NY, USA) and were described as the mean ± standard deviation (SD). An unresponsive Student’s t-test was used to compare the differences between the two groups. To test the data normality, each of the data groups was tested by a Shapiro–Wilk test. Differences between multiple groups were analyzed using a one-way analysis of variance (ANOVA) and Tukey’s post hoc test; values of *p* < 0.05 were considered statistically significant.

## 3. Results

### 3.1. Effect of NM80 and NM300 on Cell Viability

The cytotoxicity of Mg(OH)_2_ nanoparticles was confirmed. Mg(OH)_2_ nanoparticles were classified by particle size into three types: NM80 (50–100 nm), NM300 (200–400 nm), and NM700 (400–1000 nm). NM80 and NM300 did not affect the OD values compared to the control for up to 500 μg/mL (NM80; *p* = 0.17, NM300; *p* = 0.12), while NM700 administration showed a significant decrease in OD values at a concentration of 100 μg/mL (*p* = 0.0256). NM80/NM300 was used in subsequent experiments (Figure 1).

### 3.2. NM80 and NM300 Inhibit the Expression of the IL-1β Induced by LPS

The effects of NM80 and NM300 on inflammatory responses were examined. IL-1β mRNA (Figure 2A, NM80 50, 100, 250, 500 μg/mL; *p* < 0.0001, NM300 50, 100, 250, 500 μg/mL; *p* < 0.0001) and the protein (Figure 2B) expression induced by LPS were suppressed by stimulations with NM80 and NM300 in a dose-dependent manner. As shown in Figure 2C, IL-1β protein secretion induced by LPS was also inhibited by their stimulation with NM80/NM300 (*p* < 0.0001). To elucidate whether the inhibitory effect of NM80 and NM300 on IL-1β expression was dependent on changes in Mg^2+^ or pH in the culture medium, the cells were stimulated with LPS in the presence of MgSO_4_ or in the pH10.4 condition. Mg^2+^ in the culture medium (*p* = 0.97) and the pH of the culture medium (*p* = 0.99) did not affect the inhibitory effect of IL-1β expression (Figure 2D).

### 3.3. NM80 Inhibits LPS-Induced Activation of the PI3K/Akt-NF-κB Pathway

The effects of NM80/NM300 on the PI3K/Akt and NF-κB pathways were examined. The LPS-induced degradation of the IκBα protein was inhibited by its stimulation with NM80, whereas NM300 stimulation had no effect (Figure 3A). Similarly, the nuclear translocation of NF-κB (p65) protein was inhibited by its stimulation with NM80 (Figure 3B). Furthermore, LPS-induced Akt phosphorylation was inhibited by NM80 (Figure 3C). A selective inhibitor of the PI3K/Akt pathway (LY294002) inhibited Akt phosphorylation and IκBα degradation (Figure 3D). Pretreatment with LY294002 also suppressed IL-1β mRNA (*p* = 0.0005) and protein expression (Figure 3E,F). 

### 3.4. NM80 Inhibits LPS-Induced Activation of the MAPKs Pathway

The effects of NM80/NM300 on the MAPK pathway were examined. The LPS-induced phosphorylation of JNK, ERK1/2, and p38 MAPK was inhibited by NM80. NM300 only inhibited ERK1/2 phosphorylation (Figure 4A). The inhibitory effect of JNK (JNK inhibitor II), ERK1/2 (PD98059), and p38 MAPK (SB239063) was confirmed by Western blot analysis (Figure 4B). Pre-treatment with these inhibitors attenuated LPS-induced IL-1β mRNA expression (Figure 4C, JNK inhibitor II treatment; *p* = 0.03, PD98059 treatment; *p* = 0.41, SB239063 treatment; *p* = 0.005).

## 4. Discussion

Several studies have reported on the biological activity of Mg(OH)_2_ nanoparticles, including the inhibition of the *Pseudomonas aeruginosa* biofilm formation [27], the promotion of bone regeneration [28], and the inhibition of the inflammatory response in implants [24]. We found a bactericidal effect of Mg(OH)_2_ nanoparticles on dental plaque-producing bacteria, periodontopathic bacteria, and *E. coli* [29]. In this study, we investigated the effect of Mg(OH)_2_ nanoparticles on pathogen-induced IL-1β derived from periodontopathic bacteria, with the goal of using them to prevent periodontitis. 

In our previous study, NM80 outperformed NM300 and NM700 in terms of bactericidal effect [22]. We also evaluated the cytotoxicity of Mg(OH)_2_ nanoparticles on immune cells and discovered that neither NM80 nor NM300 inhibited J774.1 cell proliferation (Figure 1). We have no explanation for these contrasting results, although we believe that NM80 and NM300 are biocompatible with immune cells. However, NM700 significantly impaired the proliferation of J774.1 cells and was excluded from further studies. 

Macrophages initiate an inflammatory response following LPS stimulation. This inflammatory response activates signals from the NF-κB and MAPK pathways, resulting in the production of inflammatory cytokines such as IL-1β [12]. We demonstrated that administrating NM80 and NM300 suppressed LPS-induced IL-1β expression in J774.1 cells (Figure 2A,B). The secretion of IL-1β has been reported to be influenced by several stimuli, such as nigericin and maitotoxin, as well as *Staphylococcus aureus* infection [12]. As shown in Figure 2C, the nigericin-stimulated secretion of IL-1β in conditioned media was increased by LPS, and this enhancement was attenuated by NM80 and NM300. Thus, NM80 and NM300 have the potential to suppress IL-1β expression and extracellular secretion. 

Furthermore, magnesium ions (Mg^2+^) decrease inflammatory cytokine production [30]. However, the addition of Mg^2+^ to the culture media did not impact the expression of LPS-induced IL-1β, suggesting that Mg^2+^ from NM80 and NM300 did not reduce the inflammatory response in J774.1 cells (Figure 2D). Although the addition of NM80 and NM300 raised the pH in the conditioned media of J774.1 cells from 7.1 to 10.4, we also confirmed that these changes in the pH of the culture medium were not a major factor in the reduction in IL-1β expression by NM80 and NM300. Therefore, we speculated that the inhibition of IL-1β expression and secretion by NM80 and NM300 was dependent on the modification of the intracellular signaling activation induced by LPS. 

Several signaling molecules have been reported to be responsible for IL-1β expression. NF-κB is one of the key pathways involved in the inflammatory response. In a quiescent state, p65 and p50 are sequestered in the cytoplasm in association with IκBα. In the state of activation by several stimulants, IκBα protein can be degraded by the ubiquitin/proteasome system. The free heterodimer of p65 and p50 translocates into the nucleus and initiates the transcription of various genes, including inflammatory cytokines [31]. The administration of NM80 significantly suppressed the LPS-induced activation of the NF-κB pathway (Figure 3A,B). Previous studies have demonstrated that the PI3K/Akt pathway acts upstream of NF-κB, and the phosphorylation of Akt transactivates the NF-κB pathway [32,33]. We investigated the effect of Mg(OH)_2_ nanoparticles on PI3/Akt activation and found that the LPS-driven phosphorylation of the Akt protein was also inhibited by NM80 (Figure 3C). Interestingly, the pharmacological inhibition of the PI3/Akt cascade suppressed LPS-induced IL-1β expression via the downregulation of IκBα degradation (Figure 3D–F). These results suggest that the inhibitory effect of NM80 on IL-1β expression is dependent on the suppression of the PI3/Akt-NF-κB axis. 

Next, we focused on the MAPK pathway: another important pathway regulating the inflammatory response [34]. In the present study, NM80 inhibited the phosphorylation of JNK, ERK1/2, and p38 MAPK (Figure 4A). Furthermore, selective inhibitors of JNK, ERK1/2, and p38 MAPK downregulated the LPS-induced stimulation of IL-1β mRNA expression (Figure 4B–C. These data demonstrate that the inhibition of MAPK-mediated signaling pathways also plays an important role in the suppression of IL-1β expression by NM80. 

However, the LPS-induced phosphorylation of ERK1/2 was suppressed by NM300 but not by JNK or p38 MAPK. Unfortunately, we have no explanation for the differences in the effects of these nanoparticles on the intracellular signaling molecules involved in the inhibition of IL-1β. It has been reported that the mechanism of internalization of extracellular particles by macrophages depends on material sizes, as the proteins involved in this process are different [35]. Further studies are needed to examine correlations between the internalization process of each Mg(OH)_2_ nanoparticle and the signaling events involved in the inhibition of LPS-induced IL-1β expression. From this, it can be inferred that the molecular mechanism and the suppression of IL-1β expression by Mg(OH)_2_ nanoparticles may be due to differences in particle size. The effect of the particle size on the modification of the inflammatory response by Mg(OH)_2_ needs to be investigated in the future. Furthermore, recent studies have suggested that the application of paraprobiotics [36], probiotics [37], and postbiotics [38] can modify clinical and microbiological parameters in patients with periodontitis and peri-implant mucositis. These products should be considered in combination with Mg(OH)_2_ nanoparticles in future clinical trials and in the field of periodontics and implant dentistry.

A limitation of our present study is that the results were validated only in a monolayer culture system of cell lineage. Further studies on the effects of Mg(OH)_2_ nanoparticles on LPS-induced IL-1β expression using primary macrophages and in vivo animal models could clarify the regulatory mechanisms of inflammatory response and contribute to the development of effective treatment strategies for periodontitis.

In conclusion, our data demonstrate that Mg(OH)_2_ nanoparticles inhibit an inflammatory response when stimulated by the virulence factor of periodontopathic bacteria via the suppression of NF-κB and MAPK activation in murine macrophages. This work suggests that Mg(OH)_2_ nanoparticles may help to prevent the exacerbation of periodontitis and suppress inflammation, which occurs through the host response against the periodontal pathogen. We also found that Mg(OH)_2_ nanoparticles suppressed biofilm formation by mutant streptococci [29]. Therefore, we believe that Mg(OH)_2_ nanoparticles can be applied in dental materials to prevent infectious diseases caused by oral bacteria.

## Figures and Tables

**Figure 1 biomedicines-11-01291-f001:**
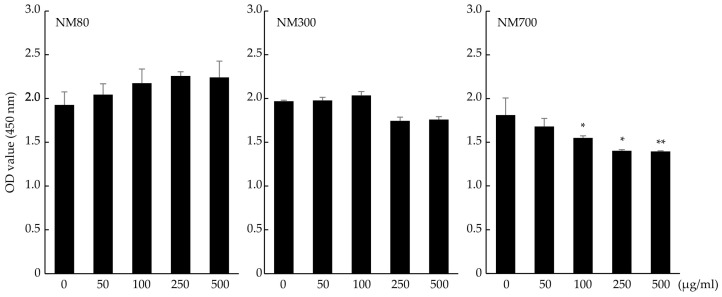
Effects of NM80, NM300, and NM700 on purification of J774.1 cells detected by the CCK-8 assay. Black bars indicate the mean value for each group, and error bars indicate the standard deviation. (* *p*< 0.05, ** *p* < 0.01).

**Figure 2 biomedicines-11-01291-f002:**
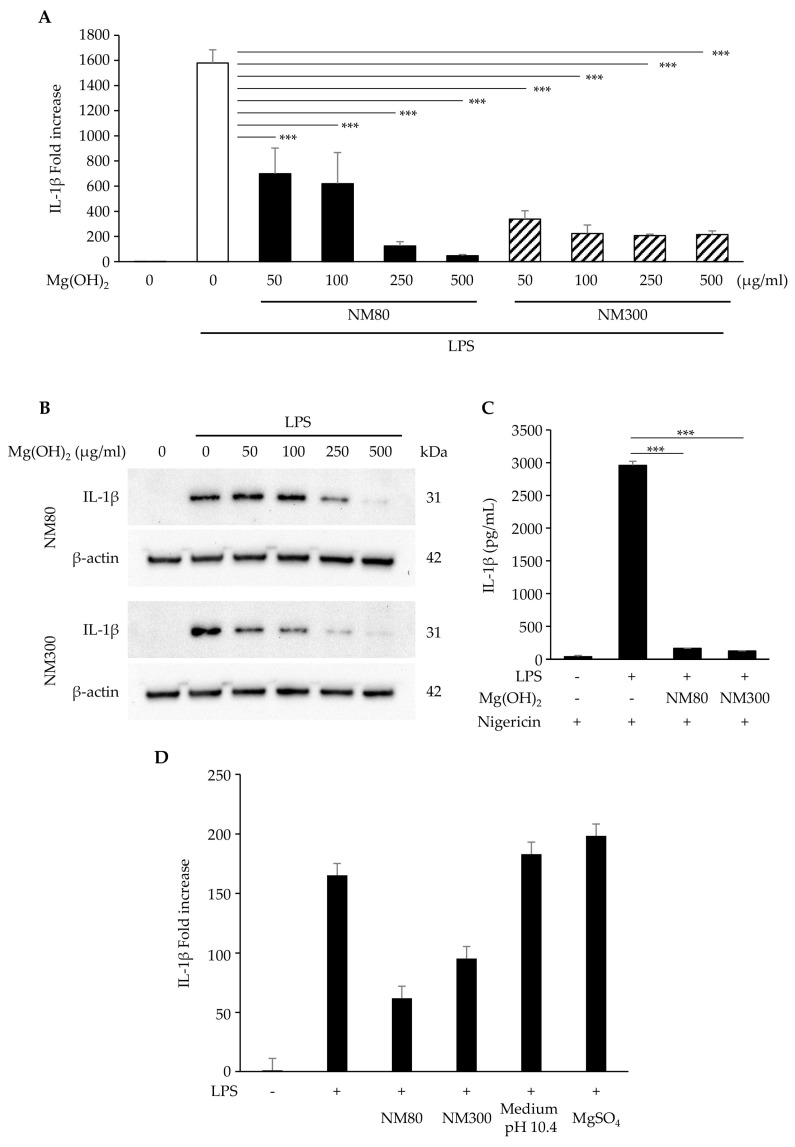
(**A**,**B**) J774.1 cells were stimulated with LPS (2 ng/mL) and NM80/NM300 for 2 h (**A**) and 6 h (**B**). (**A**) The mRNA level of IL−1β was measured using real−time RT−qPCR. (**B**) The protein level of IL−1β was detected by Western blot analysis. β−actin serves as the loading control. (**C**) J774.1 cells were treated with LPS (2 ng/mL) and NM80/NM300 (500 μg/mL) for 5 h, followed by stimulation with nigericin for 1 h. The protein secretion of IL-1β in a conditioned medium was monitored by ELISA. (**D**) J774.1 cells were exposed to NM80, NM300, and MgSO_4_ in the presence of LPS (2 ng/mL) for 2 h. In some experiments, J774.1 cells were cultured with LPS (2 ng/mL) in the culture medium adjusted to pH 10.4. The mRNA level of IL−1β was measured using real-time RT−qPCR. Black bars indicate mean values for each group, and error bars indicate standard deviations. (*** *p* < 0.001).

**Figure 3 biomedicines-11-01291-f003:**
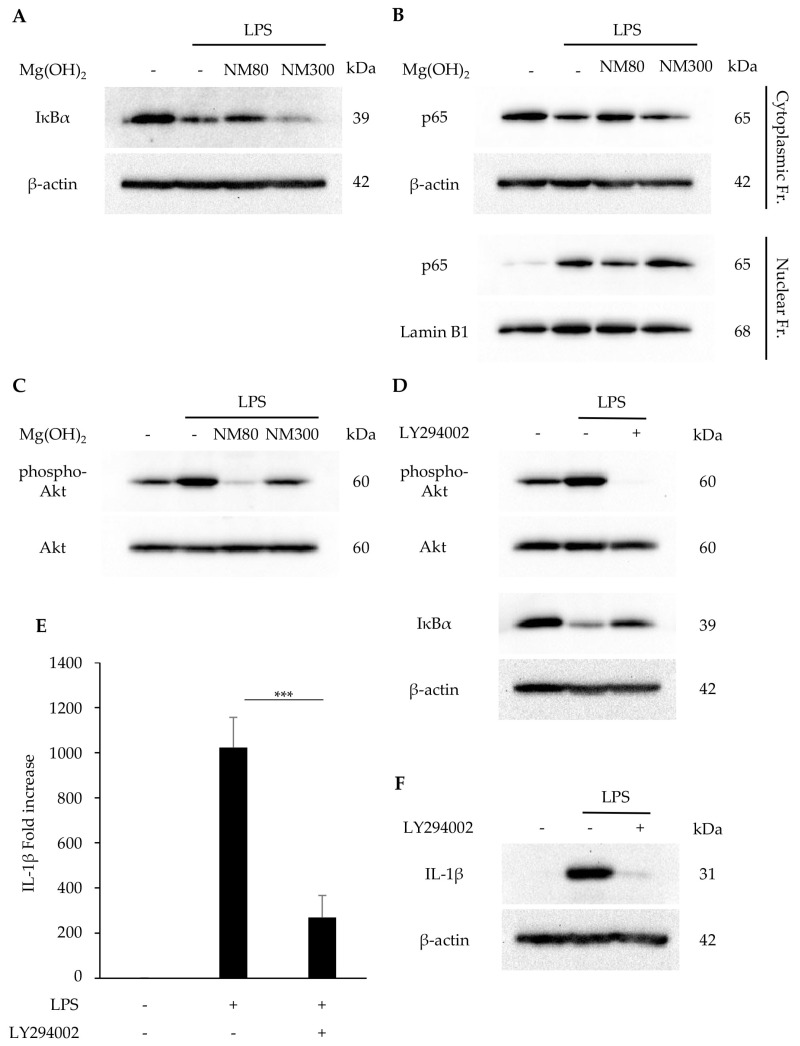
(**A**−**C**) J774.1 cells were stimulated with LPS (2 ng/mL) and NM80/NM300 (500 μg/mL) for 1 h (**A**,**B**) and 30 min (**C**). (**A**) IκBα protein was detected by Western blot analysis. β−actin served as a loading control. (**B**) NF−κB (p65) protein in cytoplasmic and nuclear fractions was detected by Western blot analysis. β−actin (cytoplasmic fractions) and lamin B1 (nuclear fractions) served as a loading control. (**C**) Phosphorylated and total Akt protein were detected by Western blot analysis. (**D**−**F**) J774.1 cells were pretreated with LY294002 for 1 h and stimulated with LPS (2 ng/mL) and NM80/NM300 for 30 min (phospho−Akt), 1 h (IκBα), 2 h (**E**), and 6 h (**F**). (**D**) Phosphorylated Akt, Akt, and IκBα proteins were detected by Western blot analysis. β−actin was used as a loading control. (**E**) IL−1β mRNA levels were measured using RT−qPCR. Black bars indicate mean values for each group and error bars indicate standard deviation (*** *p* < 0.001). (**F**) IL−1β protein was detected by Western blot analysis. β−actin served as a loading control.

**Figure 4 biomedicines-11-01291-f004:**
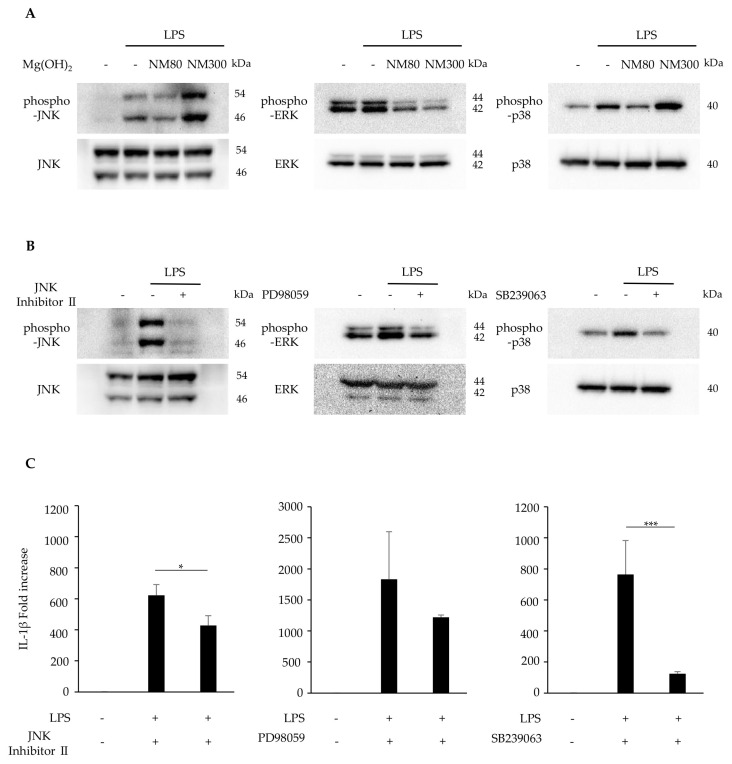
(**A**) J774.1 cells were stimulated with LPS (2 ng/mL) and NM80/NM300 (500 μg/mL) for 30 min. Protein expression of phosphorylated JNK (phospho−JNK), JNK, phosphorylated ERK1/2(phospho−ERK1/2), ERK1/2, phosphorylated p38 MAPK (phospho−p38), and p38 MAPK (p38) was detected by Western blot analysis. (**B**) J774.1 cells were pretreated with JNK Inhibitor II (50 μM), PD98059 (10 μM), and SB239063 (10 μM) for 1 h and stimulated with LPS (2 ng/mL) for 30 min. Each protein expression level was detected by Western blot analysis. (**C**) J774.1 cells were pretreated with JNK Inhibitor (50 μM), PD98059 (10 μM), SB239063 (10 μM) for 1 h and stimulated with LPS (2 ng/mL) for 2 h. The mRNA expression of IL−1β was detected by RT−qPCR analysis. Black bars indicate mean values for each group, and error bars indicate standard deviation (* *p* < 0.05, *** *p* < 0.001).

## Data Availability

The data presented in this article are available upon request from the corresponding author.

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
