# Peer review of "Mechanisms Underlying the Suppression of IL-1β Expression by Magnesium Hydroxide Nanoparticles"

_biomedicines, 2023, doi:10.3390/biomedicines11051291_

Round 1

Reviewer 1 Report

Introduction. This investigation presents information for researchers in the field of microbiology and inflammatory response in periodontal tissues. Periodontitis is a frequent chronic inflammatory diseases caused by bacterial infections. Macrophages are immune cells responsible for the rapid response to bacterial infection. The recognition of microbial pathogens, including LPS, by macrophages activates inflammatory signaling pathways and induces the production of various pro-inflammatory cytokines, such as IL-1. The production of cytokines in the periodontal tissue causes gingival attachment loss, periodontal pocket formation and alveolar bone destruction.  Recently, magnesium hydroxide has attracted attention because of its bactericidal properties.  

The aim of this study was investigated the biological effects of Mg (OH)2 on inflammatory responses induced by periodontopathic bacteria in macrophages.

The first, second and paragrah must incorporate some references. The fourth paragraph must incorporate more references.

Materials and methods.

This section showes a good structure of different subsections (i.e. reagents, cell culture, WST-8 analysis, PCR, Western blotting, ELISA), but the author must explain if this protocol is original or is based in before published experiments. Also, the authors only includes 3 references in this section, that must be increased with new references for a better scientific experimental basis.

Results

This section showes a correct structure of different subsections (cell viability, expression of the IL-1β induced by LPS, activation of the PI3K/Akt-NF-κB pathway, activation of the MAPKs pathway) and include several figures related with the experimental results.  

Discussion.

This section includes the analysis of results according the scientific evidence of previous studies.

Some references must be updated.

The authors must incorporate in this section the relevance of the results of this experimental research, and its clinical application in periodontics and implant dentistry for the control of bacteria associated to periodontitis and peri-implantitis

Conclusions. This section is not found.

References. This section must be increased with new references. Many references are older. The references include only 7 papers (30,4%) of last five years

Conclusively, the study is not ready for publication.

Reviewer 2 Report

I have read the manuscript with interest and some questions raised. Enlisted please find my comments.

Overall. General English grammar revision (Minor spelling errors).

Key words. “dentistry” and “periodontology” could be added in my opinion.

Abstract. Please add the names of the statistical tests in this section.

Introduction. Authors stated “Progressive periodontitis results in the destruction of the periodontal tissue and alveolar bone resorption, eventually resulting in tooth loss. In addition, periodontitis is associated with systemic diseases, such as diabetes, myocardial infarction, and aspiration pneumonia.”. Please add a reference for this statement.

Introduction. Authors stated “Periodontopathic bacteria such as Porphyromonas gingivalis, Tannerella forsythia, Treponema denticola, and Aggregatibacter actinomycetemcomitans (A.a) are reported to initiate and progress periodontitis”. Please add a reference for this statement.

Materials and Methods. Authors stated “Mouse macrophage cell line, namely J774.1 cells, was purchased from RIKEN CELL BANK (RCB0434) (Ibaraki, Japan) and cultured in RPMI 1640 (Wako, Osaka, Japan) containing 10% heat-inactivated fetal bovine serum (FBS) (Sigma-Aldrich), 100 units/ml penicillin, and 100 μg/ml streptomycin (Wako Pure Chemicals; Osaka, Japan).”. Please add a reference for this method.

Materials and Methods. Please add details about software used, version, Manufacturer, City and State.

Materials and Methods. Authors stated “Differences between multiple groups were analyzed using one-way ANOVA analysis of variance and Tukey's post hoc test”. ANOVA is used for gaussian distributions. Please explain how normality of data was tested.

Results. Please add P values in the text all along this section.

Discussion. Authors stated “The effect of the particle size on the modification of the inflammatory response by Mg (OH)2 should be investigated in the future”. Provide a general interpretation of the results in the context of other evidence, and implications for future research. It could be added that “Additionally, it should be considered that some recently introduced compounds have a significant influence on oral environment. The use of paraprobiotics [Butera A, Gallo S, Maiorani C, Preda C, Chiesa A, Esposito F, Pascadopoli M, et al. Management of Gingival Bleeding in Periodontal Patients with Domiciliary Use of Toothpastes Containing Hyaluronic Acid, Lactoferrin, or Paraprobiotics: A Randomized Controlled Clinical Trial. Applied Sciences. 2021; 11(18):8586], lysates [Effect of probiotic Lactobacillus rhamnosus by-products on gingival epithelial cells challenged with Porphyromonas gingivalis. Vale GC, Mayer MPA. Arch Oral Biol. 2021 Aug;128:105174..] and postbiotics [Butera A, Pascadopoli M, Pellegrini M, Gallo S, Zampetti P, Cuggia G, et al. Domiciliary Use of Chlorhexidine vs. Postbiotic Gels in Patients with Peri-Implant Mucositis: A Split-Mouth Randomized Clinical Trial. Applied Sciences. 2022; 12(6):2800. https://doi.org/10.3390/app12062800] can modify Clinical and Microbiological Parameters in periodontal patients, so also these products should be considered in combination with magnesium hydroxide in future clinical trials”. These concerns should be added to Discussion section.

Discussion. Please add a paragraph showing the limitations of the present report.

References. Some references are quite old (1996;1999;1987;1986). If possible, please switch with some more modern research. Some recent studies have been suggested in the sections above.

Round 2

Reviewer 2 Report

All comments have been answered. Thank you.

Author Response

We have carefully checked the spell and modified the manuscripts.